# MetPC: Metabolite Pipeline Consisting of Metabolite Identification and Biomarker Discovery Under the Control of Two-Dimensional FDR

**DOI:** 10.3390/metabo9050103

**Published:** 2019-05-25

**Authors:** Jaehwi Kim, Jaesik Jeong

**Affiliations:** 1Statistical Information Department, Duksung Women’s University, Seoul 01369, Korea; un7743@naver.com; 2Statistics Department, Chonnam National University, Kwangju 61186, Korea

**Keywords:** hierarchical statistical model, fdr2d, metabolite identification, latent variable, Expectation-Maximization

## Abstract

Due to the complex features of metabolomics data, the development of a unified platform, which covers preprocessing steps to data analysis, has been in high demand over the last few decades. Thus, we developed a new bioinformatics tool that includes a few of preprocessing steps and biomarker discovery procedure. For metabolite identification, we considered a hierarchical statistical model coupled with an Expectation–Maximization (EM) algorithm to take care of latent variables. For biomarker metabolite discovery, our procedure controls two-dimensional false discovery rate (fdr2d) when testing for multiple hypotheses simultaneously.

## 1. Introduction

Due to intrinsic complexity of metabolomics data, it is very challenging to preprocess the data. In essence, preprocessing of the data requires many different sequential steps including, but not limited to peak detection, peak merging, metabolite identification and peak alignment. Conducting all these steps together is not easy and each preprocessing step has been addressed separately in different studies. Especially, metabolite identification [1,2] and peak alignment [3,4,5,6,7,8] have attracted much attention compared to other preprocessing steps. Note that most preprocessing works have not taken statistical model into account to achieve their goal. But, Jeong and his colleagues employed statistical hierarchical model to address both issues such as metabolite identification and peak alignment in three different studies [1,3,4].

Once metabolomics data are preprocessed, the subsequent tasks are applied to the preprocessed data. Especially, biomarker metabolite discovery has been extensively studied [9,10,11]. Here, “biomarker metabolite” means the metabolite with statistically significant difference between two groups (e.g., cancer vs. normal group). In other words, if some metabolites have huge mean difference between groups, it is more likely that they are related somehow to cancer, implying that they could possibly play a role in cancer mechanism. In this regard, it is crucial to discover them in the scientific way because they can be used as potential targets in the course of medical treatment. From the statistician’s perspective, such issue is categorized as the multiple testing problem. When testing for multiple hypotheses simultaneously, the most important issue is how to control the false discovery rate (FDR). Benjamini and Hochberg defined FDR and introduced a stepwise test rule that uses ordered *p* values [12]. Following the seminal paper of FDR, a test rule, which controls one-dimensional false discovery rate (fdr1d), was developed [9]. However, there is a limitation that the null hypotheses corresponding to test statistics with small standard error have higher tendency of being falsely rejected, resulting in high FDR [10]. To avoid such inflation of FDR, a new test rule, which controls two-dimensional false discovery rate (fdr2d), was introduced [10]. Comparative analysis showed that the fdr2d is much better than the fdr1d in terms of the control of FDR. It is also mentioned that the FDR control is well graphically summarized in two types of plots: volcano plot and tornado plot.

Since there has been the increasing demand for the development of unified computational platform handling metabolomics data, many softwares have been developed over the last decade. However, each software has a very limited function. For example, some softwares such as XCMS Online [13] and MetSign [14] can be used for preprocessing such as peak detection and alignment while others (MetScape [15], IMPaLa [16], MetExplore [17], MetPA [18], and MetaboAnalystR 2.0 [19]) for pathway analysis and visualization. Also, MZmine2 contains some statistical analysis such as principal component analysis (PCA) and clustering. [20] The functions that each software contains are well summarized in the literature [21] (it is written in Korean and the title is translated into English by the authors). In some cases, it is important to select some meaningful information (say, biomarker) and use them for statistical analysis. Thus, we develop a bioinformatics platform that is implemented by R, which mainly consists of metabolite identification and biomarker discovery. Also, some other functions including peak merging are contained, which will be explained later in detail. In Section 2, we provide Materials and Methods. Some results are provided in Section 3, and followed by Discussion in Section 4. Additional details on the manual of the software is provided in Appendix A.

## 2. Materials and Methods

### 2.1. Metabolite Identification by a Hierarchical Statistical Model

The metabolite identification is processed by comparing sample metabolites with library metabolites. Such comparison is based on similarity measure, which is the input for our model. We consider a hierarchical statistical model for metabolite identification, which consists of four different layers. Each layer of the model, which reflect experimental situations, includes an observed or unobserved (latent) variable, explaining a part of the whole process (Figure 1 and Figure 2).

Firstly, *Y* is a binary random variable that indicates whether the metabolite in the library is in the sample (absent: 0, present: 1). In practice, we do not know what metabolites are in the sample compound and thus variable *Y* is a latent variable. Secondly, *Z* is an observable variable that indicates whether the metabolite in the library matches a particular metabolite in the sample (not matched: 0, matched: 1). As a criterion for matching, we consider the cosine score, acute angle between two mass spectra. The cosine score between two metabolites (one from sample and the other from library) is called the dissimilarity score. Matching rule is that, given a metabolite in the sample, a metabolite in the library with the smallest dissimilarity score is matched. At the end of the matching, it is possible that several sample metabolites can be matched to a metabolite in the library. Thirdly, *W* is a binary variable indicating whether or not the matching is correct (incorrect: 0, correct: 1). If the value of *Z* is zero (if the metabolite in the library does not match any metabolite in the sample), then the value of *W* does not need to be considered. Similarly, if *Y* = 0 and *Z* = 1, the value of *W* will always be zero. However, under normal circumstances, we do not know whether or not the matching is correct, implying that the variable *W* is also a latent variable. In the last layer of the hierarchy, we collect all dissimilarity scores from matched pairs (variable *S*) and then estimate the density function of the scores in the normal mixture framework. All layers of our hierarchical model are graphically summarized in Figure 2.

Since there are two latent variables, Expectation–Maximization (EM) comes in for parameter estimation. Furthermore, we consider a factorization of joint distribution function for computational simplicity because joint distribution is not easy to handle. By simple algebra, it is easily shown that the joint distribution is factored into multiplication of four marginal distributions:[Y,Z,W,S]=[Y][Z|Y][W|Y,Z][S|Y,Z,W].
Given distribution assumption for each of four variables, we then write down the complete-data log-likelihood, $L_{c}$. In E-step, we take conditional expectation of the complete-data log-likelihood given observed data, resulting in target function:$$Q\left(\theta \right)=E\left[L_{c}|Z,S\right],$$
where θ is the parameter vector. In M-step, we maximize the target function with respect to θ. More details about distribution assumption and EM are given in the previous literatures [1,3].

### 2.2. Biomarker Discovery Under the Control of Two-Dimensional FDR

Once all samples are identified, they are combined in terms of metabolites existing in all samples. After that, biomarker metabolites that have statistically significant difference between/among groups are discovered. Here, we focus on the two group problem because many problems in the real world including cancer study belong to this area. In case of classical two-sample comparison ($X_{1}$ and $X_{2}$), the null hypothesis is
$$H_{0}:\mu_{X_{1}}=\mu_{X_{2}}.$$
To test the null hypothesis, we use a typical *t*-statistic
$$T=\frac{{\bar{X}}_{1}-{\bar{X}}_{2}}{S_{p}},$$
where $S_{p}^{2}=\sqrt{S_{X_{1}}^{2}/n_{1}+S_{X_{2}}^{2}/n_{2}}$, and ${\bar{X}}_{1}$ and $S_{X_{1}}^{2}$ are sample mean and variance for group $X_{1}$. However, note that we have to test for many hypotheses simultaneously, i.e., multiple testing problem. It is well known that classical test rule does not work in multiple testing problem. Thus, we consider controlling fdr2d.

Benjamini and Hochberg introduced the concept of FDR, which is defined as the probability that null hypothesis is falsely rejected, $P(H_{0}:true|H_{0}:rejected)$ [12]. Following the seminal paper of FDR, effort of continuing improvement has been made in the mixture framework:$$\begin{array}{c} f\left(z\right)=\pi_{0}f_{0}\left(z\right)+\pi_{1}f_{1}\left(z\right), \end{array}$$
where $\pi_{0}$ and $\pi_{1}$ are the proportions of true null and alternative, respectively, and $f_{0}$ and $f_{1}$ are densities of *z* corresponding to $H_{0}$ and $H_{1}$, respectively [10,22]. One-dimensional local fdr is defined as follows:$$\begin{array}{c} \mathrm{fdr}1d=\pi_{0}\frac{f_{0}\left(z\right)}{f(z)}. \end{array}$$
For the estimation of the fdr1d, they first estimated the crucial ratio $f_{0}\left(z\right)/f\left(z\right)$ by using logistic regression in empirical Bayes sense and then combined it with ${\hat{\pi }}_{0}$ [22]. Based on the estimate of fdr1d, they developed a test rule. However, Ploner and his colleagues mentioned that Efron’s fdr1d would have some problem when making decisions for test statistics with small standard error. In other words, test statistic could be inflated due to a small standard error even when the mean difference is small, leading to the false rejection of null hypothesis. As an illustrating example, suppose that we have two hypotheses with the same observed test statistic $t_{0}$, but different standard errors. Then, the same values of $t_{0}$ are graphically represented as two distinct points (1 and 2 in Figure 3) due to different standard errors.

Clearly, they are separated in fdr2d, but not in fdr1d. In other words, the statistical decision for two hypothesis should be the same in fdr1d, but could be different in fdr2d. Thus, to reduce the inflation of the FDR, Ploner and his colleagues extended the one-dimensional FDR to two-dimensional one. The two-dimensional local fdr is defined:$$\mathrm{fdr}2d=\frac{\pi_{0}f_{0}\left(z_{1},z_{2}\right)}{f(z_{1},z_{2})},$$
where $z_{1}$ is *t* statistic and $z_{2}$ is logarithm of standard error of the test statistic. For the estimation of fdr2d, they first estimated the ratio by penalized log-likelihood approach:$$r\left(z\right)=\frac{n_{p}f_{0}\left(z\right)}{f\left(z\right)+n_{p}f_{0}\left(z\right)}$$
where $n_{p}$ is the number of permutation. And then they get the estimate of fdr2d by plugging the r(z) in the equation below:$$\mathrm{fdr}2d\left(z\right)=\pi_{0}\frac{r(z)}{n_{p}\left[1-r\left(z\right)\right]}.$$
It is mentioned that the *t* statistic reduces the combined information contained in the mean difference and the standard error to a single ratio, which is a disadvantage of Efron’s approach. In contrast, the fdr2d uses the full bivariate information, a cause of FDR reduction [10].

### 2.3. Software Implementation

#### 2.3.1. Two Major Goals

Two major goals are sequentially addressed: metabolite identification and biomarker discovery. Overall workflow of our software is graphically summarized in Figure 4.

For metabolite identification, we consider a database search algorithm. Database means a library in which information about all metabolites is known. A key idea is that to identify a metabolite in sample, we look into the library and find the metabolite with the most similar features to the sample metabolite. As a dissimilarity measure, we use cosine score:$$s=\frac{180}{\pi }cos^{-1}\left(\frac{<a,b>}{\left||a||\cdot ||b|\right|}\right),$$
where <a,b> is the inner product of two vectors and ||a|| the Euclidean norm of *a*. In our case, each component in the vector is non-negative and the range of the score is 0≤s≤90, the small cosine score meaning high similarity.

For biomarker discovery, we need to reduce the dimension of the data such that a scalar value is assigned to each metabolite. Two popular reductions, which transform mass spectrum for each metabolite to a scalar value, are widely used: base peak chromatogram (the maximum value of the intensity) or total ion chromatogram (sum of all intensities). After dimension reduction, extra data manipulation such as log-transformation and scaling is taken into account. More specifically, after log-transformation, scaling within each column is performed, resulting in relative proportion (i.e., sum to 1). Thus, input for biomarker discovery is data matrix from two different groups. Each row represents each metabolite and each column each subject. Multiple testing is performed on data matrix under the control of fdr2d.

#### 2.3.2. Kernel Density Estimator

In the last layer of the hierarchy, we estimate the density function of the score in the finite mixture framework. We fit two or three component normal mixture to the scores depending on the situation. In comparison to a parametric approach, we also consider kernel density estimator (KDE):$$\hat{f}\left(t\right)=\frac{1}{n}\sum_{i=1}^{n}K_{h}\left(t-s_{i}\right)=\frac{1}{nh}\sum_{i=1}^{n}K\left(\frac{t-s_{i}}{h}\right),$$
where h>0 is a bandwidth and *K* is a non-negative density function. Note that, even though optimal bandwidth is used as a default, bandwidth can be selected by users.

Some built-in plots are added to the package to give a chance to graphically check the accuracy of the parametric density estimation. In case of parameters estimated (say, estimates of mean, variance, and component proportion), we provide trace plot to check the convergence of EM algorithm. Comparing the parametric density estimator to the KDE, the accuracy of the parameter estimation can be further checked.

## 3. Results

### 3.1. Identification

#### 3.1.1. Data

A mixture of 35 amino acids, fatty acids and organic acids were prepared in pyridine. All GCxGC/TOF-MS analyses were performed on a LECO Pegasus 4D time-of-flight mass spectrometer (TOF-MS) with a Gerstel MPS2 auto-sampler. The Pegasus 4D GCxGC/TOF-MS instrument was equipped with an Agilent 6890 gas chromatograph featuring a LECO two stage cryogenic modulator and secondary oven. Experimental details are provided in the Appendix A of the original literature [3].

The experiments were conducted with temperature gradient of 5 ${}^{\circ }$C/min with ten replicates. We selected two of them for illustration (the Dataset1 in Experiment I in original paper). For identification, we used those two data sets obtained from the same compound mixture. One of them is used as sample and the other one library. Before conducting identification, we performed peak merging (the peak with the biggest area is selected), resulting in 76 peaks.

#### 3.1.2. Identification Results

As a criterion for identification, we consider confidence measure, which is defined as a posterior probability that a metabolite in library exists in sample given dissimilarity score
$$P_{j}=P\left(Y_{j}=1|Z_{j},S_{j};\hat{\theta }\right),$$
where $\hat{\theta }$ is parameter estimates and $S_{j}$ is the score assigned to the *j*-th metabolite in library.

As a result of identification, we have a table containing two columns such as name and confidence measure. Figure 5 shows the top 20 metabolites in terms of posterior probabilities in the descending order. Among all metabolites in the library, those 20 metabolites are more likely to exist in the sample. Complete lists including all identification results is provided in Appendix A.

Regarding parameter estimation, Figure 6 provides the trace plot of four different parameter estimates by EM. Based on the plots, it is clear that Expectation–Maximization (EM) algorithm converges fast within the first 200 iterations for all cases.

For density estimation, we consider two different approaches: parametric vs. non-parametric (kernel density estimator). Two density estimates are provided in different colors in Figure 7 (three component normal mixture in red and kernel density estimator in black).

Clearly, the parametric estimate overlaps the kernel density estimate very well, implying that EM-algorithm works well (i.e., mean and variance of each component are appropriately estimated).

### 3.2. Real Data Anlaysis

#### 3.2.1. Schisandra Data

For biomarker metabolite discovery, we consider a new data set consisting of 57 schisandra chinesis from two different countries; 27 species of China and 30 species of Korean Schizandra chinensis are purchased, dried, and crushed to powder. The produced powder is extracted and then underwent derivatization to be analyzed by the GC/MS. Experimental details are given in the literature [23]. Before statistical analysis, the data are statistically processed: centering and log-transformation, resulting in 3226 by 57 data matrix.

#### 3.2.2. Biomarker Discovery

Since the data come from two different countries (Korea and China), the classical two-sample comparison is considered. We have 3226 null hypotheses to be tested simultaneously (a hypothesis for each metabolite). Controlling two-dimensional fdr, we find some metabolites showing significant difference between two countries. For this, we borrow some code from *OCplus* R package developed by Ploner and his colleagues.

As suggested by Ploner and his colleagues, biomarker discovery results are graphically represented as volcano plot and tornado plot. In both plots, the *x*-axis is mean difference. The *y*-axis, however, is different. That is, the log(se) is used in the volcano plot while -log(*p* value) is used in the tornado plot.

In Figure 8, each point in the plot represents a hypothesis for a metabolite and each isoline corresponds to each level of FDR. The test rule is that, given a FDR level, all points outside the corresponding isoline are rejected. For example, at the fdr2d level of 0.05, 1092 null hypothesis are rejected. Furthermore, 1326, 1782, and 2120 hypothesis are rejected at the level of 0.1, 0.2, and 0.3, respectively.

## 4. Discussion

Our ultimate goal is to build a streamlined bioinformatics tool that can be easily used to handle omics type big data, ranging from preprocessing to various statistical analysis. As a first step, two major issues are considered: metabolite identification and biomarker metabolite discovery. It should be emphasized that, for metabolite identification, both the sample and the library have to be provided from users, i.e., no built-in library in the program. The reason is that the composition of library heavily depends on the type of sample analyzed.

As a future direction, classification (group assignment) using the discovered biomarkers will be added to the pipeline. Network analysis (metabolite–metabolite interaction) would be another direction. From different point of view, the tool will be extended to the multi-group problem. The current version of the tool is now available at the github website (https://github.com/jjs3098/CNU-Bioinformatics-Lab). If there is an update, we then make it available as soon as possible. For reproduction of all plots provided in this study, we provide detailed explanations about how to reproduce those plots in the Appendix A.

## Figures and Tables

**Figure 1 metabolites-09-00103-f001:**
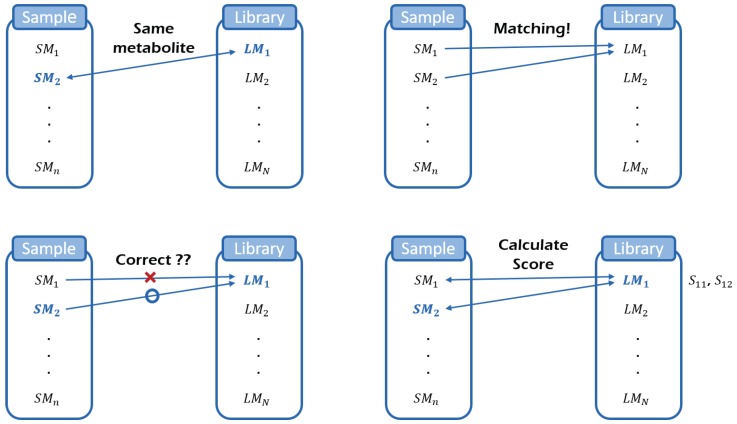
Schematic representation of 4 variables. *Y* is a binary latent variable representing the existence of each library metabolite in sample (**top left**). *Z* is a binary variable (observable) representing one of two status: “matched (Z=1)” or “not matched (Z=0)” (**top right**). *W* (**bottom left**) is a binary latent variable representing the matching results: correct (W=1) or not correct (W=0). *S* is the similarity score between two mass spectra (**bottom right**).

**Figure 2 metabolites-09-00103-f002:**
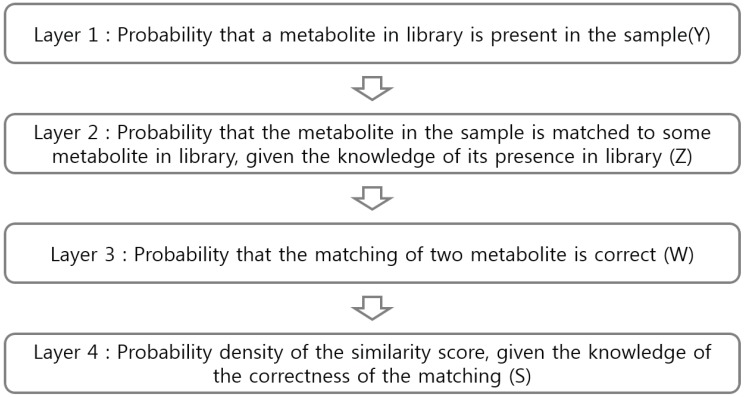
Graphical representation of the hierarchy of our model.

**Figure 3 metabolites-09-00103-f003:**
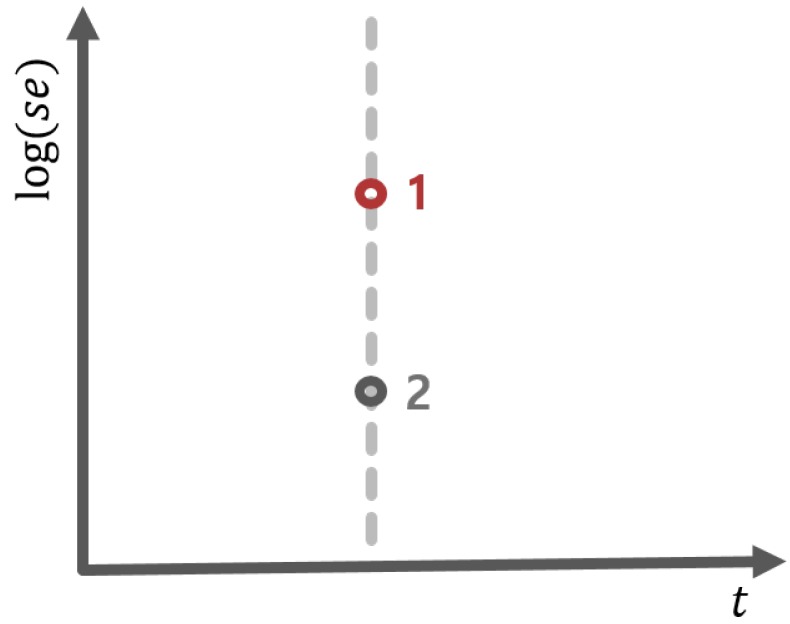
Limitation on Efron’s FDR. Two points have the same value of $t_{0}$. However, the standard error of point 1 (in red) is larger than that of point 2 (in grey).

**Figure 4 metabolites-09-00103-f004:**
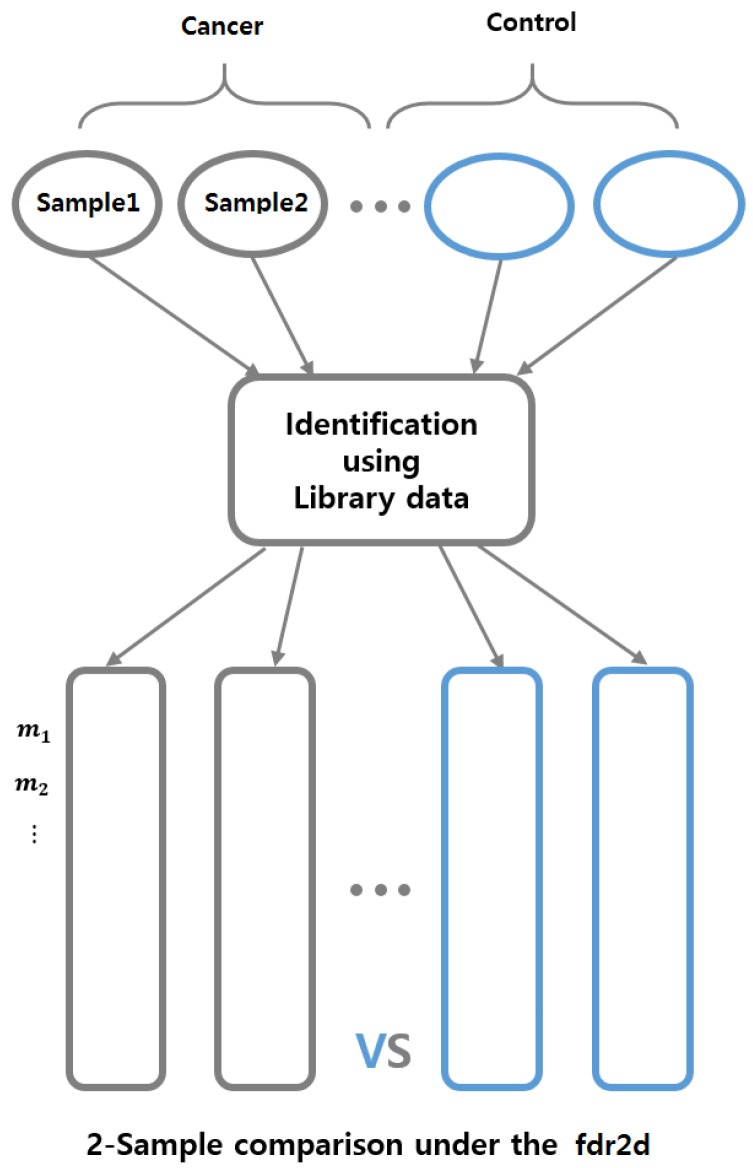
Process of metabolite identification and biomarker discovery.

**Figure 5 metabolites-09-00103-f005:**
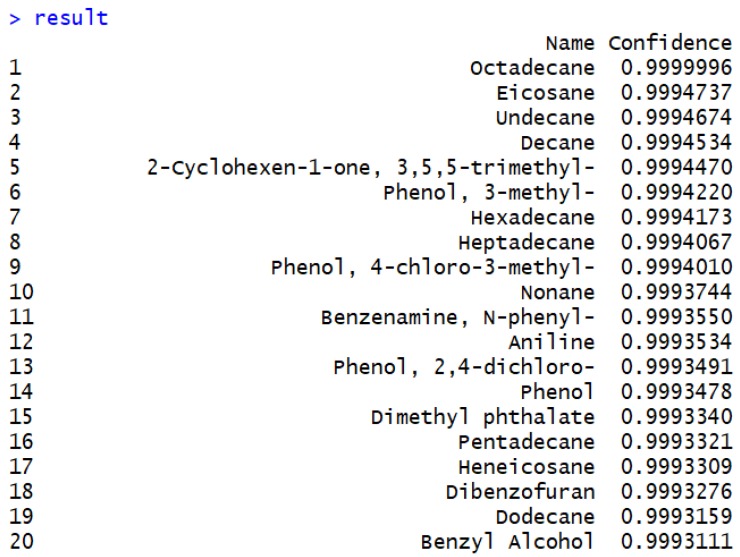
Results of metabolite identification.

**Figure 6 metabolites-09-00103-f006:**
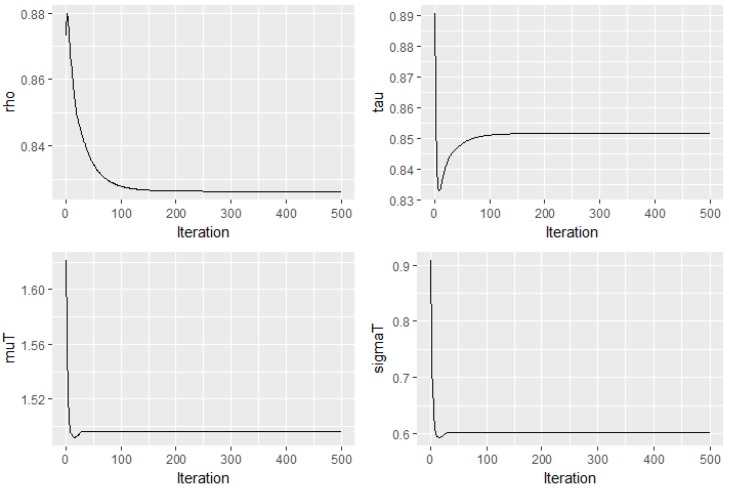
Trace plot of four parameter estimates by EM-algorithm. Parameter ρ (**top left**), parameter τ (**top right**), parameter $\mu_{T}$ (**bottom left**), and parameter $\sigma_{T}^{2}$ (**bottom right**).

**Figure 7 metabolites-09-00103-f007:**
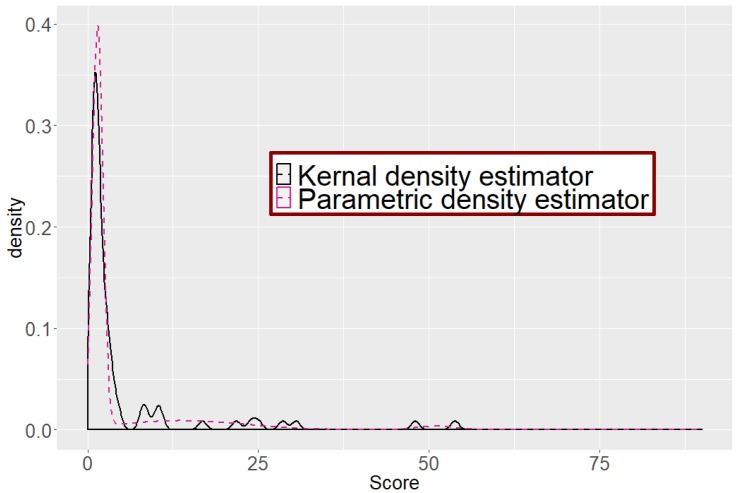
Parametric vs. Non-parametric density estimator. Parametric density estimator (three component normal mixture) is represented in red and non-parametric density estimator (kernel density estimator) is in black. The parametric density estimator overlaps well the non-parametric density estimator.

**Figure 8 metabolites-09-00103-f008:**
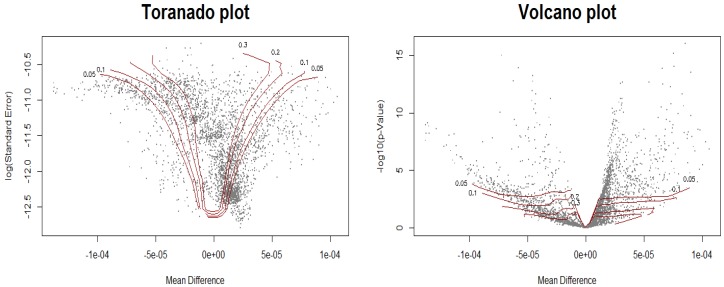
Tornado plot (**left**) vs. volcano plot (**right**). Four isolines corresponding to the four different levels of FDR, i.e., (0.05, 0.1, 0.2, 0.3) are represented in red.

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
