# Peer review of "MetPC: Metabolite Pipeline Consisting of Metabolite Identification and Biomarker Discovery Under the Control of Two-Dimensional FDR"

_metabolites, 2019, doi:10.3390/metabo9050103_

Reviewer 1 Report

The study describes two interesting methods useful to solve two main issues in metabolomics: metabolite annotation/identification and multiple testing. Moreover, an R-implementation is provided. Some major points should be addressed.

1- The method described for metabolite identification is very interesting. Some comments should be added about the following points:

1.1- can the method be applied to HNMR? or only to mass spectrometry?

1.2- why are not retention time and cross-section included in the similarity function?

1.3- is it correct to use 'identification' or it is better to use 'annotation'? In case of annotation, which is the level (I suppose 1=idenfication or 2)?

1.4- which is the format to use for the library and for the metabolite representation?

2- The results section should be improved. I suggest to better describe the data sets (type of data, aim of the study, number of samples, structure of the data set), and to report the data analysis as a tutorial using the R-package developed. For example: the input data have the structure..., then we have applied...finding...the library have the structure...and so on. The reader should be able to follow the steps described in the theoretical part using the data.

3- The results of 2d-fdr should be compared with those obtained by 1d-fdr. 

4- In the practical section, it should be described how the R-package developed can be intergrated in XCMS or in a generic workflow used for metabolomics from raw data to biomarkers discovery. The format of the data in input and the output should be described to allow the reader to design his workflow.

Minor points

1- Please replace 'work' in the introduction with 'study' or 'investigation' depending on the cases.

2- Some english errors are present, check the english form.

3- Re-arrange the materials as: introduction, materials and methods, results, discussion.

4- The dissimilarity function is the angle distance, I suggest to use it instead of cosine score or the authors should justify the use of the term 'cosine score'.

Reviewer 2 Report

Manuscript presented by Kim and Jeong is supposed to cover a very important topic in metabolomics research but because of a very poor presentation, lack of appropriate description of examples, lack of conclusions and very short and unclear discussion it is hard to see how is this manuscript and indeed the tool contributing. I would suggest that authors completely rewrite the manuscript, actually show how it can be used on real data (with appropriate description) and resubmit this new publication. As is although it appears that paper contributes a possibly useful analysis tool it is very hard to understand its value.

Author Response

Round  2

Reviewer 1 Report

The authors satisfactorily addressed all the comments. Some minor points should be solved:

1- check the numbering of the references, the correspondence number-reference and re-numbering if needed;

2- re-write sentences in rows 48-50 in light of the new structure of the manuscript;

3- use 1d-fdr or fdr1d and 2d-fdr or fdr2d in the text (for example,in row 84 authors use both 1d-fdr and fdr1d);

4- use Supplementary Material instead of Supplementary Data;

5- I did not find Supplementary Data II.

Reviewer 2 Report

I would like to thank authors for their changes and improvement to the manuscript. Overall I think that now this an interesting paper presenting possibly useful application. At the same time some of the claims made in the paper are exaggerated as authors for example claim that this tool provides combined preprocessing and analysis while at the same time in the examples mass spectrometry data was pre-processed with MetaboAnalysis 2.0 and MZmine toolbox which both actually provide preprocessing and analysis (and this was not mentioned  in the introduction). Still statistical analysis for peak assignments presented in the paper and provided by the R library is interesting for MS metabolomics applications and could perhaps be expended for NMR applications as well. I would suggest to authors to stress that aspect of the contribution, without presenting it as an all inclusive tool and instead show comparison of metabolite assignment provided with their approach with for example assignment of metabolites methods provided by MZmine or another tool. Presenting benefits of this method would be of interest and worthy of publication in my opinion. Additionally, as authors present some novel datasets it would be very important to provide them within the GITHUB as examples and for further development of methodologies.

In addition paper's accessibility to readers will greatly benefit with further correction of English language. 

Author Response

Round  3

Reviewer 2 Report

I would like to thank the authors for making requested changes and providing their code and data for further investigations. In its current form this paper and the software tool will likely be of interest to other researchers.